# Dimethylglyoxime Clathrate as Ligand Derived Nitrogen-Doped Carbon-Supported Nano-Metal Particles as Catalysts for Oxygen Reduction Reaction

**DOI:** 10.3390/nano11051329

**Published:** 2021-05-18

**Authors:** Luping Xu, Zhongqin Guo, Hanyu Jiang, Siyu Xu, Juanli Ma, Mi Hu, Jiemei Yu, Fengqi Zhao, Taizhong Huang

**Affiliations:** 1Science and Technology on Combustion and Explosion Laboratory, Xi’an Modern Chemistry Research Institute, Xi’an 710065, China; sungirlxu5@163.com (L.X.); jianghanyu612@126.com (H.J.); xusy99@163.com (S.X.); mjl_821113@sohu.com (J.M.); amphite@hotmail.com (M.H.); 2Shandong Provincial Key Laboratory of Fluorine Chemistry and Chemical Materials, School of Chemistry and Chemical Engineering, University of Jinan, Jinan 250022, China; 17862918959@139.com (Z.G.); chm_yujm@ujn.edu.cn (J.Y.)

**Keywords:** dimethylglyoxime, metal nano-particles, nitrogen-doped carbon, catalysts, oxygen re-duction reaction

## Abstract

Nitrogen-doped carbon-supported metal nano-particles show great promise as high-performance catalysts for novel energies, organic synthesis, environmental protection, and other fields. The synergistic effect between nitrogen-doped carbon and metal nano-particles enhances the catalytic properties. Thus, how to effectively combine nitrogen-doped carbon with metal nano-particles is a crucial factor for the synthesis of novel catalysts. In this paper, we report on a facile method to prepare nitrogen-doped carbon-supported metal nano-particles by using dimethylgly-oxime as ligand. The nano-particles of Pd, Ni, Cu, and Fe were successfully prepared by the pyrolysis of the corresponding clathrate of ions and dimethylglyoxime. The ligand of dimethylglyoxime is adopted as the source for the nitrogen-doped carbon. The nano-structure of the prepared Pd, Ni, Cu, and Fe particles are confirmed by X-ray diffraction, scanning electron microscopy, and trans-mission electron microscopy tests. The catalytic performances of the obtained metal nano-particles for oxygen reduction reaction (ORR) are investigated by cyclic voltammetry, Tafel, linear sweeping voltammetry, rotating disc electrode, rotating ring disc electrode, and other technologies. Results show that the nitrogen-doped carbon-supported metal nano-particles can be highly efficient catalysts for ORR. The results of the paper exhibit a facile methodology to prepare nitrogen-doped carbon-supported metal nano-particles.

## 1. Introduction

Clean and sustainable energies and technologies, such as water splitting cells, metal-air batteries, and fuel cells, are in great demand due to their merits of high-efficiency, high energy intensity, fast start-up, environmental friendliness, and low operating temperature [1,2]. At present, the carbon-supported metal based nano-particles, such as Pt, Pd, etc., are the major catalysts for fuel cells, which assure the proceeding work of fuel cells. For example, carbon-supported platinum (Pt)-based nano-particles are the state-of-the-art ORR catalysts [3,4]. However, the shortcomings of high-cost, easily-poisoned, and poor-durability have become obstacles to their large-scale applications [5]. Therefore, developing high-performance and low-cost electrocatalysts for ORR is urgent for the widespread applications of fuel cells. Recent studies show that various transition metal based materials, such as single atom catalysis [6], metal-organic frameworks (MOFs)-derived transition metal sulfide [7], oxide and nitride, etc. [8,9], show great promise to be high-performance catalysts for novel energies. With the deepening understanding of the transition metal-based catalysts, transition metal based nano-particles exhibit great potential to be high performance catalysts for many fields, such as ORR, HER, oxy-gen evolution reaction, etc. In the procedure of transition metal-based catalyst preparation, how to control the size and morphology of metal nano-particles is the key technology.

The RhSi-based nano-particles has reported as catalysts for hydrogen evolution rea-ction (HER) [10]. In all kinds of methodologies to prepare metal nano-particles, adopting MOFs as intermediates is one of the most feasible methods. MOF-derived nitrogen-doped carbon-based functional materials have been reported as high-performance ORR catalysts due to their unique physical and chemical properties such as large specific surface area, polymetallic sites, etc. [11].

Co-CoO/N-rGO and Ni-NiO/N-rGO were synthesized via a simple annealing of graphene oxide-supported cobalt and nickel salts, respectively [12]. On the other hand, the transitional metal (Fe, Co, Ni) nanoparticles encapsulated in nitrogen-doped carbon nanotubes (M/N-CNTs, M = Fe, Co, Ni) were also synthesized via a solid-state thermal reaction [13]. To prevent the oxidation of metal nano-particles, nano-carbon-based support was derived by the pyrolysis of an organic ligand. The derived carbon can also enhance the conductivity and play a synergistic role in catalysis. Except the synthesis of carbon-supported elemental substances, metal oxides were also obtained from MOFs. For example, Cu_2_O nanoparticles were prepared by a facile methodology, which was adopted as catalysts for ORR [14]. The nano-sized CoFe_2_O_4_ also showed high catalytic performance for OER [15]. The nano-sized CoFe_2_O_4_ was obtained from the pyrolysis of prussian blue analog MOFs. The catalytic performance of benchmark Pt-based alloys for ORR could also be improved by tuning particle structures [16]. To modulate the structures of different elemental nano-particles, different ligands have been investigated. Zeolitic imidazolate frameworks (ZIFs) [17] provided materials based on simple zeolite structures using different imidazole analogues. 2-Methylimidazolate is a typical ligand for ZIF-8, which exhibits an interesting nanopore topology formed by four-ring and six-ring ZnN_4_ clusters [18]. Trifluoroacetic acid/acetic acid based MOFs were synthesized to enhance CO_2_ adsorption and catalytic properties [19]. 1H, 5H-benzo(1,2-d,4,5-d) bistriazole was also adopted as a ligand to prepare FeN_3_-based catalysts for the reduction of CO [20]. The synthesis and application of MOFs in electrochemistry fields had been reviewed [11], which clearly pointed out that the MOFs were excellent candidates for a large number of applications. The MOF-based materials have been widely studied and kinds of organic materials have been reported as ligands for MOFs. Until now, the MOFs are usually made from ligands with high price and complex structure. How to simplify the synthesis procedure and decrease the cost of MOFs have attracted great attention. This paper adopts dimethylglyoxime clathrate as the ligand to prepare nitrogen-doped carbon-supported transition metal-based nano-particles as catalysts for ORR. Dimethylglyoxime clathrate has the special characteristics of low decomposition temperature and simple procedure to form coordinate materials. Based on the special characteristics, we select the dimethylglyoxime clathrate as the ligand to prepare nitrogen-doped carbon-supported metal nano-particles as catalysts for ORR.

In this paper, we reported the synthesis of nitrogen-doped carbon-supported metal nano-particles of Fe, Ni, Cu, and Pd from the pyrolysis of the clathrate of corresponding metal ions and dimethylglyoxime (dmgH_2_) at 350 °C. The structures of the nitrogen-doped carbon-supported metal nano-particles were confirmed by scanning electron microscopy, transmission electron microscopy, and other technologies. As a supplement, the catalysis of the nitrogen-doped carbon-supported Fe, Ni, Cu, and Pd nano-particles for ORR was also investigated. Results showed that the nano-particles have good catalytic performance for ORR. The results of the paper provide a novel way to prepare metal nano-particles at low temperature by the pyrolysis of metal-dmgH_2_ clathrate.

## 2. Experimental Section

### 2.1. Materials Synthesis

Graphene oxide (GO) solution was prepared from natural graphite by a modified Hummers’ method, which has been reported before [21]. The dimethylglyoxime, nickel chloride Hexahydrate (NiCl_2_·6H_2_O), Cupric Chloride Dihydrate (CuCl_2_·2H_2_O), Ferric Chloride Hexahydrate (FeCl_3_·6H_2_O), Potassium Hydroxide (KOH), Nafion solution (20 wt%) and ethanol were obtained from Sinopharm Chemical Reagent Co., Ltd (Shanghai, China). The high-purity argon and oxygen gas were purchased from Baode Gas Co Ltd (Jinan, China) Except for the GO solution, the other reagents were analytic purity grade and used without any treatments.

The typical synthesis procedure of nitrogen-doped metal nano-particles is illustrated in Scheme 1, and the obtained materials are named as M@NC.

The typical synthesis process of Ni@NC is as follows: 1.0015 g dmgH_2_ was completely dissolved in 100 mL ethanol, and then 2 mL GO (6 g/L) solution was added and stirred until evenly dispersed. Then 1.0303 g nickel chloride hexahydrate (NiCl_2_·6H_2_O) was added to the solution by stirring until completely dissolved. The solution turns bright red. The mixed solution was poured into a round bottom flask and the pH adjusted to 8 with NaOH solution. The finally obtained solution was heated and stirred at 85 °C for 30 minutes, then reacted for 2 h at 55 °C. The obtained precipitation was washed with ethanol and de-ionized water. The finally obtained red precipitate was (dmg)_2_Ni. Finally, the dried (dmg)_2_Ni was heated in Ar at 350 °C for 2 h and the obtained black powder was nitrogen-doped carbon-supported Ni nano-particles named as Ni@NC. The temperature was determined according to the TG-DSC test results that displayed in Scheme 1. The TG-DSC tests were conducted by a STA449F3 equipment of Netzsch Co. Ltd (Selb, Gemany). It could be seen from Scheme 1 that the pyrolysis temperature of the coordinate of dimethylglyoxime with metals is about 300 °C. To avoid the growth of metal nano-particles, we selected 350 °C as the final synthesis temperature. The preparation procedure of Fe-, Cu-, and Pd-(dmg)_2_ was quite similar to that of (dmg)_2_Ni and the obtained materials are named as Fe@NC, Cu@NC, and Pd@NC, respectively. Ravi et al. had reported the synthesis of FeCo alloy that was encapsulated by nitrogen-doped carbon nanotube by heating melamine at 950 °C [22].

### 2.2. Structure Characterization

A Bruker D8 advanced X-Ray diffractometer (XRD) (Bruke DXS, Karlsruhe, German) with Cu-Kα (λ = 1.5418 Å) was used to collect the XRD data between 10° and 80° (2θ). The images of the transmission electron microscopy (TEM), high-resolution TEM (HRTEM) were obtained with a JEOL 2010 unit (Tokyo, Japan). A Perkin Elmer PHI5300 spectrometer with monochromatized Mg Kα radiation (Waltham, MA, USA) was employed to detect the X-ray photoelectron spectroscopy (XPS) of the catalysts.

### 2.3. Electrochemical Measurements

A mixed solution of 5.0 mg electrocatalysts Ni-NC, 450 μL DI water and 50 μL Nafion solution (20% in weight) was ultrasonicated for 40 min to form a well-dispersed ink. Then, 5 μL of the ink was pipetted onto the surface of the glass-carbon electrode for CV and RDE tests, while 8 μL of the ink was pipetted onto the RRDE test electrode.

The electrochemical tests were conducted by a CHI 760E electrochemical workstation, which is a product of China Shanghai Chenhua Co. Ltd., with a three-electrode cell setup, which included glassy carbon disk (working electrode), carbon electrode (counter electrode), and Ag/AgCl (reference electrode). The cyclic voltammetry (CV), linear sweep voltammetry (LSV), electrochemical impedance spectroscopy (EIS), and Tafel tests were conducted. The CV tests of the materials were conducted in both Ar- and oxygen-saturated 0.1 M KOH electrolyte between 0.2 and −0.8 V (vs. Ag/AgCl) with the sweeping rate of 0.005 V s^−1^. The LSV, Tafel and EIS tests were also conducted in an O_2_-saturated electrolyte. A rotating ring disc electrode (RRDE) and rotating disc electrode (RDE) tests were evaluated at different rotating speeds by a RRDE 3A electrode that combined with a CHI 760E electrochemical workstation. For the RDE test, CV tests were conducted in Ar- and O_2_-saturated electrolyte to activate the materials. After that, for the RDE, the LSV tests were conducted at the sweeping rate of 0.005 mV/s with the rotating speed ranged from 625 rpm to 2500 rpm. The RRDE tests were conducted at rotating speeds of 1600 rpm with the same sweeping rate. Based on the RDE tests, the electron transfer number (*n*) of ORR can be calculated according to the Koutechy–Levich equation (Equation (1)) [23]:(1)$$\frac{1}{J}=\frac{1}{J_{L}}+\frac{1}{J_{K}}=\frac{1}{B\omega^{-1/2}}+\frac{1}{J_{K}}\;$$
The slope of the *K-L* line could be written as Equation (2) [7]:(2)$$B=0.62nFC_{0}{\left(D_{0}\right)}^{2/3}v^{-1/6}$$
where *n* is the electron transfer number, *F* is the Faraday constant (96,485 C/mol), $D_{0}$ is the diffusion coefficient of O_2_ in the electrolyte, v is the kinetic viscosity, and $C_{0}$ is the bulk concentration of O_2_ in the electrolyte. The constant 0.62 is adopted when the rotating speed *ω* is expressed in rad/s.

Based on the RRDE tests, the *n* and percentage ratio of hydrogen peroxide (*H*_2_*O*_2_*%*) on the electrode surface could be calculated according to Equations (3) and (4) [24]:(3)$$n=\frac{4\times I_{d}}{I_{d}+I_{r}/N}\;$$
(4)$$H_{2}O_{2}\%=200\times \frac{I_{r}/N}{I_{d}+I_{r}/N}$$
where the value of *N* is 0.39.

## 3. Result and Discussions

The structure of the catalysts were firstly investigated by XRD and XPS tests and the results are displayed in Figure 1.

The XRD patterns in Figure 1a clearly revealed that the diffraction peaks of Cu-NC were corresponding to the indexed Joint Committee on Powder Diffraction Standards (JCPDS) no. 78-2076 of Cu with micro amount of CuO [14]. The XRD patterns of Ni-NC and Pd-NC were indexed to corresponding to no. 04-1043 and 46-1043, which confirmed the successful synthesis of Ni and Pd nano-particles, respectively. However, to the Fe-NC, no obvious peaks could be indexed, which should be attributed to the Fe particles’ size being too small to produce effective diffraction.

To investigate the state of the compositional elements, the X-ray photoelectron spectroscopy (XPS) tests of each catalyst were conducted and the full spectra of the XPS of Ni-, Cu-, Pd-, and Fe-NC are shown in Figure 1b, which clearly proved the successful synthesis of the corresponding materials. The signal of C and N in Figure 1b should be attributed to the dmgH_2_ derived carbon.

Figure 1c shows the high-resolution XPS of Pd element. The two peaks corresponding to 340.59 and 335.30 eV should belong to the Pd 3d_3/2_ and 3d_5/2_ [25], which also affirmed the successful synthesis of Pd nano-particles. The existence of Pd^2+^ could be attributed to the partially-oxidized Pd [26], which may be happened during the process of samples preparation for XPS tests. Figure 1d shows the high-resolution XPS of Ni. The two peaks corresponding to 872.43 and 852.74 eV should belong to Ni 2p_1/2_ and 2p_3/2_, respectively [27]. The existence of Ni^2+^ could be attributed to the NiO [28]. Figure 1e shows the high-resolution XPS of Fe. The two peaks with the binding energy 727.75 and 713.75 eV should be attributed to the 2p_1/2_ and 2p_3/2_, respectively [29]. The existence of Fe^2+^ could be attributed to the partial oxidized Fe. This clearly proved the successful synthesis of Fe from the coordinate of (dmg)_2_Fe.

Figure 1f shows the high-resolution XPS of Cu. The peaks with the binding energy of 954.85 and 934.73 eV should belong to the 2p_1/2_ and 2p_3/2_ of Cu [30], and the peaks of 959.31 and 943.58 eV should belong to the 2p_1/2_ and 2p_3/2_ of CuO [31]. The XPS test results of Pd-NC, Ni-NC, Cu-NC, and Fe-NC are consistent with the results of XRD tests. The deconvoluted high-resolution XPS of C and N of the materials are quite similar. The typical XPS of C 1s of Pd-NC is shown in Figure 1g. It clearly shows the existence of C=O (287.9 eV), C=N (285.5 eV) and C=C/C–C (284.59 eV) bonds [32]. Figure 1h shows the high-resolution XPS of N 1s. The four peaks spectra in the spectra can be deconvoluted into oxygenated-N (403.59 eV), graphitic-N (400.60 eV), pyrrolic-N (398.99 eV), and pyridinic-N (398.69 eV) [33]. The combined N-C bond proved the successful synthesis of the nitrogen-doped carbon support. Graphitic-N and pyridinic-N are beneficial to the occurrence of ORR. The pyridinic-N has a significant effect on electron delocalization in the carbon framework and, hence, have a profound effect on the ORR performance [34]. The morphology and corresponding elemental mapping of the synthesized materials are displayed in Figure 2.

Figure 2 shows the SEM and corresponding element mapping of the (dmg)_2_-metal-derived materials. The morphologies of Pd-NC (a), Ni-NC (b), Fe-NC (c), Cu-NC (d) showed that the obtained metal nano-particles evenly distributed on the support, which should be the reduced graphene oxide [35]. The corresponding metal of Pd, Ni, Cu, and Fe should be derived from the clathrate of (dmg)_2_Pd, (dmg)_2_Ni, (dmg)_2_Cu, and (dmg)_2_Fe. From all the SEM of the obtained materials, it could be seen that the distribution of carbon, nitrogen and metal are consistent with each other, which proved the successful synthesis of the corresponding materials. This result is also consistent with the XRD test results. On the other hand, the ligand-derived nitrogen-doped carbon also prevents the aggregation of derived metal particles, which eventually result in the even distribution of metal particles on the support.

The HRTEM images of Cu-NC, Fe-NC, Ni-NC, and Pd-NC are shown in Figure 3. The crystal lattice spacing of 0.25 nm, 0.194 nm and 0.173 nm in Figure 3A could correspond to the (111), (200), and (220) facets of Cu. The lattice fringes in Figure 3B confirm the existence of iron rather than amorphous structure. Different lattice distances corresponding to different facets of iron are detected. On the other hand, the SAED patterns also similar to the results of non-crystal materials, which should be attributed to the existence of different lattice facets. The disappearance of obvious XRD patterns should be attributed to the diameter of the particles are too small that make the diffraction of X-ray difficult. In Figure 3C, the crystal lattice spacing of 0.174 nm and 0.206 nm could be attributed to the (200) and (111) facets of Ni. The corresponding SAED patterns also proved the crystal structure of Ni particles [36]. The HRTEM images of Pd in Figure 3D illustrates the crystal lattice spacing of 0.195 nm and 0.239 nm, which matches well with the (111) and (200) facets of Pd. Similar structure was also observed in the research of Pd based catalysts for ORR [37]. The HRTEM results are coincided with the XRD test results. Catalysts with a similar structure were also reported by Vorobyeva et al. [38].

First, the catalytic performances of the materials for ORR were examined by cyclic voltammetry (CV) tests. Figure 4a showed the CV tests of the derived materials at a sweeping rate of 0.005 V/s in both Ar- and O_2_-saturated 0.1 M KOH electrolyte. Some obvious peaks were clearly observed in the Figure 4a, which should be attributed to the happening of ORR on the electrode surface. In contrast, it is clearly shown that the onset potential and peak current intensities of ORR of the catalysts are quite different. This should be determined by the intrinsic performance of the derived materials. It is clearly shown that the onset potential of Pd-NC is higher than that of other materials.

To further ascertain the catalytic performance of the materials for ORR, the LSV tests of all the materials are conducted and the results are showed in Figure 4b. The onset potential (*E_onset_*) and half potential (*E*_1/2_) of Pd-NC catalyzed ORR are 0.95V, and 0.87 V (vs. RHE), respectively, which is the highest in all the prepared materials. It is also clearly shown that the *E_onset_* and *E*_1/2_ of the derived Cu-NC are 0.93, 0.87 V, that of the Ni-NC are 0.91, 0.84 V, and that of Fe-NC are 0.86, 0.77 V, respectively. Compared with Pt/C catalyst, the onset potential of Pd-NC is only 0.1 V lower than that of Pt/C catalysts (1.05 V). On the other hand, it is also observed that the limiting current intensity of Pd-NC is about 0.987 mA cm^−2^, which surpasses that of the Pt/C (Pt 5 wt%) catalyst. The limiting current of the Cu-NC, Ni-NC and Fe-NC catalysts are 0.283, 0.457, and 0.177 mA cm^−2^, respectively.

Figure 4c showed the Tafel tests of all the obtained materials. The Tafel slopes of Fe-NC, Ni-NC, and Cu-NC were 233.37, 252.19, and 295.36 mV dec^−1^, respectively. Thus, the Tafel slope of Pd-NC was 218.87 mV dec^−1^, and a low Tafel slope indicates a lower polarization for the reaction of ORR [39].

Figure 4d showed the electrochemical impedance spectroscopy (EIS) tests of all the catalysts for ORR. The inset in Figure 4d is the modulated equivalent circuit. R_0_ is the ohmic resistant, which mainly composed by the ohmic resistant and solution resistance between the working electrode and the reference electrode [40]. R_1_ and R_2_ represent the reaction resistant, which are usually attributed the direct four-electron reaction and two-electron reaction of ORR. The values of R_0_, R_1_, and R_2_ of each catalyst are displayed in Table 1.

Table 1 clearly showed that the resistant of Pd-NC was the lowest, which should be attributed to the high catalytic performance of Pd-NC for ORR. The ohmic resistance of R_1_ and R_2_ should be resulted from the four-electron and two-electron pathways of ORR in the alkaline electrolyte. To investigate the catalytic mechanism of the obtained materials for ORR, the RDE and RRDE tests of the obtained Pd-NC, Ni-NC, Fe-NC, and Cu-NC in oxygen saturated 0.1 M KOH electrolyte are shown in Figure 5, Figure 6, Figure 7 and Figure 8, respectively.

A comparison of the RDE tests of Cu-NC, Ni-NC, Pd-NC and Fe-NC (Figure 5a, Figure 6a, Figure 7a and Figure 8a) showed that, of all the materials, the current intensity increased with increasing rotating speed of the electrode, which should be attributed to the enhanced diffusion rate and shortened diffusion distance of oxygen on the electrode surface [41]. Yu Zhou et al. revealed that the meso-Fe−N−C electrocatalyst with the open porous enables the maximal exposure of highly active for ORR. The coordination between the Fe and nitrogen enhances the catalytic activity and stability for ORR [42].

The *K-L* line in Figure 5b, Figure 6b, Figure 7b and Figure 8b showed the *n* of the obtained materials’ catalyzed ORR. It is clearly showed that the value of *n* quite approach to 4. This means that the ORR major happened through the four-electron pathway. The nitrogen-doped carbon-supported metal nano-particles have good catalytic performance for ORR. The ratio of two-electrons on the electrode surface is rather low. A similar phenomena was also observed in the research on nano-cobalt-based catalysts for ORR [43]. The catalytic mechanism of the catalysts for ORR was also investigated by RRDE tests and the results are shown in Figure 5c, Figure 6c, Figure 7c and Figure 8c. All of the diagrams showed that the disc current intensities were much higher than that of the ring electrode. This could be attributed to the high catalytic performance of the materials.

The dependence of the n and $H_{2}O_{2}\%$ on the electrode voltage of Cu-NC, Ni-NC, Pd-NC, and Fe-NC catalyzed ORR are illustrated in Figure 5d, Figure 6d, Figure 7d and Figure 8d, respectively. It is clearly shown that, to the catalysts, the *n* approach 4 and the percentage of $H_{2}O_{2}\%$ approached 0. This result is quite consistent with the results of the RDE tests. The high catalytic performance should be attributed to the synergistic effect of the metal nano-particles and the nitrogen-doped carbon [44]. The strong chemical attachment and electrical coupling between the electrocatalytic nano-particles and nanocarbon improves the activity and durability of nonprecious metal-based electrocatalysts for ORR [45]. To investigate the catalytic performance of different alloys, we displayed the catalytic performances of some metals in comparison in Table 2.

Table 2 clearly shows that the catalytic performance of Pd-NC is superior to some transition metal-based catalysts despite there also being some other catalysts with higher performances. Compared with the other metal-based catalysts, the preparation temperature of this kind of catalyst is the lowest.

## 4. Conclusions

In this paper, we report a facile method to prepare nitrogen-doped carbon-supported metal nano-particle-based catalysts for ORR by the pyrolysis of dimethylglyoxime-metal ion clathrate. The dimethylglyoxime is a low-cost ligand for the preparation of MOFs that can be easily loaded onto graphene oxide or other supports. The structure of the obtained materials were examined by XRD, XPS, and HRTEM tests, which proved the successfully synthesized Pd-NC, Fe-NC, Ni-NC, and Cu-NC materials. The catalytic performance of the materials for ORR were examined by CV, LSV, Tafel, EIS, RDE, and RRDE tests, which clearly showed that the all the catalysts showed excellent performance for ORR. On the whole, this paper reported a facile method to prepare nitrogen-doped metal nano-particles as high-performance catalysts. The dimethylglyoxime is an effective ligand to prepare metal nano-particles.

## Data Availability

Data available on request due to restrictions eg privacy or ethical.

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
