# Peer review of "Dimethylglyoxime Clathrate as Ligand Derived Nitrogen-Doped Carbon-Supported Nano-Metal Particles as Catalysts for Oxygen Reduction Reaction"

_nanomaterials, 2021, doi:10.3390/nano11051329_

Round 1

Reviewer 1 Report

The article "Dimethylglyoxime clathrate as ligand derived nitrogen-doped carbon supported nano-metal particles as catalysts for oxygen reduction reaction" written by a team of Chinese scientists describes the preparation of doped carbon nanoparticles and the study of their catalytic properties. The presence of an organic ligand caused the nanoparticles protected from oxidation. The authors chose dimethylglyoxime as ligands, however, the rationale for the choice of which is not given. The atoms of the metals Pd, Ni, Cu and Fe were chosen as the coordination centers, as they are probably the most catalytically active. Authors should move all calculation formulas and theoretical information from the results discussion part to the experimental part. In general, the obtained part of the results is small and it does not describe important scientific results, rather routine measurements of electrocatalytic constants and parameters, the authors do not even provide an adequate comparison of the data obtained with known analogs to indicate efficiency. The work should be seriously revised and improved so that it can meet the quality and level of Nanomaterials Journal.

Author Response

Authors’ Response:

Thanks for the reviewer’s constructive comments to our manuscript. It could be seen that the reviewer is an expert of catalysts for oxygen reduction reactions. We have revised the paper according to the comments.

Page 5 Line 2:

Until now, the MOFs are usually made from ligands with high price and complex structure. How to simplify the synthesis procedure and decrease the cost of MOFs have attracted great attention. This paper adopts the dimethylglyoxime clathrate as ligand to prepare nitrogen doped carbon supported transition metal based nano-particles as catalysts for ORR. The dimethylglyoxime clathrate has the special characteristics of low decomposition temperature and simple procedure to form coordinate materials. Based on the special characteristics, we select the dimethylglyoxime clathrate as ligand to prepare nitrogen doped carbon supported metal nano-particles as catalysts for ORR.

Page 7 Line 10:

Ravi et al had ever reported the synthesis of FeCo alloy that capsulated by nitrogen-doped carbon nanotube by heating melamine at 950 ℃ [22].

Reference 22

[22] Nandan R, Pandey P, Gautam A, Bisen OY, Chattopadhyay K, Titirici MM, et al. Atomic Arrangement Modulation in CoFe Nanoparticles Encapsulated in N-Doped Carbon Nanostructures for Efficient Oxygen Reduction Reaction. ACS Appl Mater Interfaces. 2021;13:3771-81.

Reviewer 2 Report

In this article, the authors fabricated nitrogen-doped carbon supported metal nano-particles by using dimethylglyoxime as a ligand and use them as electrocatalyst for ORR. The synthesis and characterizations are well performed and results are convincing. This work can be published in Nanomaterials after few minor corrections.

  1. Author should include some discussion on the synthetic temp. Most of the report on similar field consider 800-900 degree to synthesize N-doped carbon and NPs@NC. However, the author used 350 deg. as synthesis temp. Why? Did author compare the result with atleast one at 800 or 900 deg. temp.
  2. Please add a table describing the best activity among the 4 metal ions and best value from literature  as  a comparison.
  3. What is the role of N on NC on the ORR. I find an interesting paper on this in ACS Applied Materials & Interfaces 12 (40), 44689-44699 discussing the effect of different N on ORR.
  4. Is it possible to deduce the coordination of NPs to any sort of N especially.

Author Response

In this article, the authors fabricated nitrogen-doped carbon supported metal nano-particles by using dimethylglyoxime as a ligand and use them as electrocatalyst for ORR. The synthesis and characterizations are well performed and results are convincing. This work can be published in Nanomaterials after few minor corrections.

  1. Author should include some discussion on the synthetic temp. Most of the report on similar field consider 800-900 degree to synthesize N-doped carbon and NPs@NC. However, the author used 350 deg. as synthesis temp. Why? Did author compare the result with at least one at 800 or 900 deg. temp.

Authors’ Response: Thanks for the reviewer’s constructive comments. It could be know that the reviewer is an expert in the field of novel catalysts for oxygen reduction reaction. We have revised the manuscript according to the comments and the FeCo based catalysts for ORR of reference 22th is compared with our results.

Page 7 Line 3:

The temperature is determined according to the TG-DSC test results. It could be seen from Schematic 1 that the pyrolysis temperature of the coordinate of dimethylglyoxime with metals is at 300 ℃. To avoid the growth of metal nano-particles, we selected 350 ℃ as synthesis temperature.

Page 7 Line 10:

Ravi et al had ever reported the synthesis of FeCo alloy that capsulated by nitrogen-doped carbon nanotube by heating melamine at 950 ℃ [22].

Reference 22

[22] Nandan R, Pandey P, Gautam A, Bisen OY, Chattopadhyay K, Titirici MM, et al. Atomic Arrangement Modulation in CoFe Nanoparticles Encapsulated in N-Doped Carbon Nanostructures for Efficient Oxygen Reduction Reaction. ACS Appl Mater Interfaces. 2021;13:3771-81.

  1. Please add a table describing the best activity among the 4 metal ions and best value from literature as a comparison.

Authors’ Response: Thanks for the reviewer’s constructive comments. We have compared our results with the others’ and the data are displayed in Table 2.

Page 21 Line 5:

To investigate the catalytic performance of different alloys, we displayed the catalytic performances of some metals in Table 2.

Table 2 Comparison of the catalytic performance of Pd with other metals.

Metal

E0 (V)

Ip (mA/cm2)

Reference

Pd

0.90

6.0

This work

Zn-N4

0.905

6.1

[46]

Co-N-C

0.87

5.4

[47]

Ti-O

0.80

5.0

[48]

Table 2 clearly showed that the catalytic performance of Pd-NC is superior to some transition metal based catalysts despite there are also some other catalyst has higher performances. Compared with the other metal based catalysts, the preparation temperature of this kind catalyst is the lowest.

References

[46] Jiang R, Chen X, Liu W, Wang T, Qi D, Zhi Q, et al. Atomic Zn Sites on N and S Codoped Biomass-Derived Graphene for a High-Efficiency Oxygen Reduction Reaction in both Acidic and Alkaline Electrolytes. ACS Appl En Mater. 2021;4:2481-8.

[47] Yang S, Yu Y, Dou M, Zhang Z, Wang F. Edge-functionalized polyphthalocyanine networks with high oxygen reduction reaction activity. J Am Chem Soc. 2020;142:17524-30.

[48] Chisaka M, Xiang R, Maruyama S, Daiguji H. Efficient phosphorus doping into the surface oxide layers on TiN to enhance oxygen reduction reaction activity in acidic media. ACS Appl En Mater. 2020;3:9866-76.

  1. What is the role of N on NC on the ORR. I find an interesting paper on this in ACS Applied Materials & Interfaces 12 (40), 44689-44699 discussing the effect of different N on ORR.

Response: Thanks for the reviewer’s careful remind. We have compared the role of nitrogen with different combining states on ORR. The results of the mentioned paper (Reference 34) were compared with that of the manuscript.

Page 20 Line 5:

The pyridinic-N has a significant effect on electron delocalization in the carbon framework and hence have a profound effect on the ORR performance [34].

Reference

[34] Jena HS, Krishnaraj C, Parwaiz S, Lecoeuvre F, Schmidt J, Pradhan D, et al. Illustrating the role of quaternary-N of BINOL covalent triazine-based frameworks in oxygen reduction and hydrogen evolution reactions. ACS Appl Mater Interfaces. 2020;12:44689-99.

  1. Is it possible to deduce the coordination of NPs to any sort of N especially.

Authors’ Response: Thanks for the reviewer’s constructive comments. We have discussed the coordination of Fe NPs to nitrogen.

Page 20 Line 5:

Yu Zhou et al revealed that the meso-Fe−N−C electrocatalyst with the open porous enables the maximal exposure of highly active for ORR. And the coordination between the Fe and nitrogen enhances the catalytic activity and stability for ORR [42].

Reference

[42] Zhou Y, Yu Y, Ma D, Foucher AC, Xiong L, Zhang J, et al. Atomic Fe Dispersed Hierarchical Mesoporous Fe–N–C Nanostructures for an Efficient Oxygen Reduction Reaction. Acs Catal. 2020;11:74-81.

Round 2

Reviewer 1 Report

The authors made the necessary changes to the text of the article and answered questions of interest to the reviewers. This is especially true for the presentation of both introduction and results and discussion parts and interpretation of the data obtained and the relevance of the work in general. The general view of the work has become clearer. The manuscript can be published in Journal.